# Classification of Diffuse Glioma Subtype from Clinical-Grade Pathological Images Using Deep Transfer Learning

**DOI:** 10.3390/s21103500

**Published:** 2021-05-17

**Authors:** Sanghyuk Im, Jonghwan Hyeon, Eunyoung Rha, Janghyeon Lee, Ho-Jin Choi, Yuchae Jung, Tae-Jung Kim

**Affiliations:** 1Department of Neurosurgery, College of Medicine, The Catholic University of Korea, Seoul 06591, Korea; imshns@hanmail.net; 2School of Computing, Korea Advanced Institute of Science and Technology, Daejeon 34141, Korea; jonghwanhyeon@kaist.ac.kr (J.H.); janghyeon5@gmail.com (J.L.); hojinc@kaist.ac.kr (H.-J.C.); ycjung7@kaist.ac.kr (Y.J.); 3Department of Plastic and Reconstructive Surgery, College of Medicine, The Catholic University of Korea, Seoul 06591, Korea; reyrha@catholic.ac.kr; 4Department of Hospital Pathology, Yeouido St. Mary’s Hospital, College of Medicine, The Catholic University of Korea, Seoul 06591, Korea

**Keywords:** digital pathology, deep transfer learning, convolutional neural network, oligodendroglial tumor, glioma

## Abstract

Diffuse gliomas are the most common primary brain tumors and they vary considerably in their morphology, location, genetic alterations, and response to therapy. In 2016, the World Health Organization (WHO) provided new guidelines for making an integrated diagnosis that incorporates both morphologic and molecular features to diffuse gliomas. In this study, we demonstrate how deep learning approaches can be used for an automatic classification of glioma subtypes and grading using whole-slide images that were obtained from routine clinical practice. A deep transfer learning method using the ResNet50V2 model was trained to classify subtypes and grades of diffuse gliomas according to the WHO’s new 2016 classification. The balanced accuracy of the diffuse glioma subtype classification model with majority voting was 0.8727. These results highlight an emerging role of deep learning in the future practice of pathologic diagnosis.

## 1. Introduction

Gliomas are the most common brain tumours that are believed to derive from neuroglial stem cells. On the basis of their histological features, they have been classified as astrocytic, oligodendroglial, or ependymal tumours, and have been assigned World Health Organization (WHO) grades I to IV, which represent the malignant degrees [1]. Huge progress in genetic profiling in brain tumor has recently led to changes in classification and treatment [2]. Therefore, the new (2016) WHO classification of tumors of the central nervous system ends the era of traditional diagnostic approaches that are based on histologic criteria only and incorporates molecular biomarkers [3]. Over 75% of the diffuse gliomas in adults are astrocytic. Oligodendroglial tumors account for less than 10% of the diffuse gliomas [4].

The classification of glioma subtype is a key diagnostic process, because the available treatment options, including conventional chemotherapy and targeted therapies, differ between Astrocytoma, Glioblastoma, and oligodendroglioma (ODG) patients. As for gliomas, prominent examples include the Isocitrate Dehydrogenase 1 (IDH1) mutation in diffuse gliomas [5,6], O6-methylguanine–DNA methyltransferase (MGMT) promoter methylation status in Glioblastomas [7] and 1p/19q codeletion in ODGs [8,9]. Especially, the 1p 19q codeletion is the genetic hallmark of ODGs with IDH1/IDH2 mutation [10,11]. The codeletion of chromosomal arms 1p 19q is a characteristic and early genetic event in ODGs, and patients with 1p 19q codeleted tumors showed a better prognosis, increased survival, and enhanced response to chemotherapy. Information on the 1p 19q status is a useful diagnostic assessment in morphologically challenging cases to substantiate the diagnosis of an ODG [12].

Histologic typing and the grading of diffuse gliomas is challenging task for pathologists, because tumor cell diversity of gliomas make it difficult to discriminate in precise microscopic criteria. This tendency resulted in a high rate of interobserver variation in the diagnosis of diffuse glioma, including oligodendroglioma (ODG) [13]. AI-based automated solution is a must-have system for the accurate diagnosis of diffuse glioma for pathologists. Pathology has a long history of artificial intelligence (AI) as much as any other field of medicine. For example, digital pathology is a system that digitizes glass slides into binary files, and then analyzes pathology information with AI algorithms. For the analysis of pathology images, an AI algorithm, such as deep convolutional neural network, has been used for the detection of tumor cells, classification of tumor subtype, and diagnosis of disease [14,15].

The deep learning based convolutional neural network (CNN) model has recently shown high performance in the field of image classification and object detection. To achieve high performance, a large amount of training dataset is required for training CNN-based deep learning model. However, it is difficult to collect a large amount of datasets in the clinical domain, and they usually have an imbalanced data problem between the disease-positive and disease-negative groups. To solve this problem, we used the transfer learning method for classifying gliomas subtypes and the grading of diffuse gliomas. As far as we know, this is the first study for deep learning aided classification of diffuse gliomas using real world digital pathology images that are generated from routine clinical practice, specifically in the cases according to the updated 2016 WHO classification for diffuse glioma.

## 2. Materials and Methods

### 2.1. Dataset and Image Curation

Whole slide images were obtained using hematoxylin and eosin-stained slides that were scanned from the 468 gliomas collected at the Catholic University of Korea Yeouido St.Mary’s Hospital from 2017 to 2019 during routine clinical 1p/19q fluorescence in situ hybridization (FISH) test. Figure 1 illustrates the study design and dataset selection. All of the cases had pathology data, including the diagnosis and grades (grade II, III, and IV) according to 2016 WHO classification. Among them, full molecular profiles could be assessed in 369 cases, which included IDH1 mutation status that was tested by immunohistochemistry, and 1p/19q co-deletion status that was examined by FISH. Based on the above molecular result, WSIs from IDH mutant, 1p/19q co-deleted ODGs (N = 38), and 1p/19q codeletion (–) diffuse glioma subtypes (non-ODG glioma) (N = 331) were used for developing the ODG and non-ODG glioma binary classification model (Table 1). Additionally, WSIs from the 468 diffuse glioma dataset were used in binary classification of diffuse gliomas into grade II–III and grade IV. Each model was performed separately. The clinical information was obtained from an electric medical record. The Institutional Review Board of St. Mary’s Hospital approved and reviewed this work (SC18RNSI0005).

The whole slide digital pathology images were generated during the performance of the 1p/19q FISH test using the automated BioView Duet scanning system (BioView, Rehovot, Israel). The images from diffuse glioma cohorts were reviewed to remove images having tissue process artifacts, including a section fold, pen mark, poor staining, and bubbles. Representative region of interest (ROI) images containing brain tumor were manually selected for each WSI that passed a quality control review [16] during routine clinical practice. 37,548 ROI images and 152,535 ROI images are prepared for deep transfer learning of the classification of ODG and diffuse glioma grading, respectively.

### 2.2. The Preprocessing of Dataset

37,548 ROI images (40× objective magnification with 1024 horizontal and 1024 vertical RGB pixels) for the ODG classification dataset (N = 369) were allotted into three groups: training (27,041 ROIs, N = 132), validation (3006 ROIs, N = 18), and testing (7501 ROIs, N = 219) sets for model training and testing using the knapsack algorithm [17,18]. With a total of 38 ODG positive patients, 9461 ROI images from 19 patients were used for training and 2617 from 16 patients were used for testing model. The total ROI images were separated into three groups as a 7:1:2 ratio, training (70%), validation (10%), and test set (20%) as shown Table 2.

The model was trained, validated, and tested with 224 × 224 pixel sized patches, which were obtained from non-overlapping ‘patches’ from ROI images. This resulted in tens of patches per ROIs, depending on the original size, and we filtered out the tiles containing more than 50% background in random sampling.

Another dataset was prepared for conducting transfer learning to classify gliomas into grade II, III, and grade IV. The 152,535 ROI images from 468 WSIs of diffuse gliomas with grading according to the WHO’s 2016 classification were prepared for this analysis. In Table 3, we allotted them into three groups as training (N = 293), validation (N = 81), and testing (N = 94) sets using the knapsack algorithm [17,18]. In the grade IV gliomas, 201 patients were used for training and 67 grade IV gliomas were used for testing the model. The rest of settings were used in the same way as described above.

### 2.3. The Process of Random Sampling Patches from ROI Images

The cropped ROI images patches (224 × 224 pixels) were selected to adjust the input image size for the pre-trained ResNet50V2 model. Randomly selected patches were used for training, validation, and testing ResNet50V2 model. During random patch selection, if the patch was filled with more than 50% of white background, then the patch was discarded and a new patch was selected again (Figure 2). We repeated the above process *n* times to obtain *n* patches for training.

### 2.4. Random Image Augmentation

Rather than using an input image as it is, the image augmentation technique produces several different images from the given input image by applying various tasks, such as randomly zooming, stretching, distorting, adjusting brightness, adding noise, and changing colors. When augmenting the patches, several image operations were probabilistically applied, as specified in the following Table 4. The probability of each operation applied was determined independently.

### 2.5. The Process of Transfer Learning Using ResNet50V2

For model selection, we compared ResNet50V2 [19], Inception V4 [20], Xception [21], and DenseNet201 [22] by performing three epochs of training and then evaluated with validation dataset. First, we selected ResNet50V2 as base model for transfer learning, because it showed the highest performance among the four pre-trained models. Second, we freeze the weights of the selected model to prevent an optimizer from optimizing the weights of the model during training process. Third, remove the original last fully connected (FC) layer, which classifies an arbitrary input image into 1000 classes and then add a new fully-connected layer to classify glioma subtype. Finally, train the modified model carefully with our own glioma image data. The new layer is only trained, because the rest of the layers are frozen, except for the newly added layer.

For one given input image, eight patches were randomly selected in each epoch and trained, validated, and tested. We used TensorFlow 2 [23] and two NVIDIA Tesla V100 GPUs for training and the batch size was 2048. The Stochastic Gradient Descent [24,25] with momentum [26] algorithms were used as an optimizer, the learning rate was 0.001, and exponential decay $e^{-0.1*\mathrm{epoch}}$ was applied [27]. Because the dataset was unbalanced, we oversampled the ODG images, so that the ratio of ODG and non-ODG glioma images used for training was 1:1. Nineteen epochs of training were performed with the above settings.

### 2.6. Grading Diffuse Gliomas

In this supplementary experiment, we chose MnasNet, EfficientNet-B4, EfficientNet-B5, and DenseNet201 [22] as the base models for transfer learning. We perform transfer learning for each model using the dataset that we prepared for this supplementary experiment. Furthermore, we used the early stopping technique based on the validation dataset to prevent the overfitting of each model. As a result, the following validation loss values were identified for each model, as in Table 5.

## 3. Results

A total 38 ODG and 331 non-ODG glioma patient images were used for training, validating, and testing the ResNet50V2 model for the classification of the glioma subtype. We achieved a balanced accuracy of 0.8727 using the majority voting technique. The strategy of deep transfer learning and the major voting technique is illustrated in Figure 3.

For the supplementary experiment, a total of 148 patients for low-graded Oligodendroglial tumor and 320 patients for high-graded Oligodendroglial tumor were used for training, validating, and testing. Among the four models we used in this experiment, DenseNet201 achieved an accuracy of 0.6810 and balanced accuracy of 0.5678.

### 3.1. Application of Random Image Augmentation for the Prevention of Overfitting

Deep convolutional neural networks require large amounts of images in the training stage, but, due to the nature of the medical data, the number of ODG tumor images is relatively small when compared to non-ODG glioma. If training is performed with this small number of ODG images without further mitigation, then it is highly likely that the convolutional neural network will be overfitted or not converge, even if it finished the training. Therefore, to solve this problem, this study uses the random image augmentation technique [28], as shown in Figure 4.

### 3.2. Deep Transfer Learning Framework for the Analysis of Pathology Images

We used the CNN-based model that is pre-trained on ImageNet [29], and their optimized parameters were transferred to our glioma pathology image dataset by the freezing method. ResNet50V2 was selected by comparing the training loss among the pre-trained models, such as ResNet50V2 [19], Inception V4 [20], Xception [21], and DenseNet201 [22] (Table 6). We used ResNet50V2 as a base model for transfer learning and utilized pathology images from glioma patients for the classification of the glioma subtype. In this experiment, we also used transfer learning in the same manner as described above to various models such as MnasNet [30], EfficientNet [31] and DenseNet201 [22].

### 3.3. The Performance of Classification Model Using Majority Voting

Based on the computational strategy that is outlined in Figure 3 and Figure 4, we present the two main results in Table 7.

First, we performed classification with the ResNet50V2 model for distinguishing ODG and non-ODG gliomas. The balanced accuracy was 0.7870 and F1 score was 0.7231 (Table 7). Second, we develop classification models that classify whole slide images into ODG positive and negative using the majority voting method. Majority voting was performed with the predicted labels for each patient’s pathological image to determine the final negative/positive label of the patient. Subsequently, the performance was evaluated by comparing the actual image labeling with the predicted labels in each patient. The balanced accuracy of our model was 0.8727, which is higher than previous results [14] and comparable with the results from pathologists.

### 3.4. The Performance of Diffuse Glioma Grading Model

In this experiment, we performed the binary classification of grade II, III, and grade IV diffuse glioma images using transfer learning of four CNN based classification models. Subsequently, we compared the classification performance of each model, as in Table 8. The metrics showed accuracy, precision, recall, and balanced accuracy. Because the dataset that we used in this experiment is also imbalanced, we used the balanced accuracy as a key performance metric. As a result, we achieved the highest balanced accuracy of 0.5801 on MnasNet.

## 4. Discussion

Recent advances in genetics in brain tumors have provided profound insights into the biology of gliomas, associating specific genetic aberrations with histopathological classification. Such changes provide critical information regarding the outcomes of the ODG patient treated with radiotherapy and adjuvant chemotheapy when compared to non-1p/19q codeleted glioma [32]. However, in the view point of testing and diagnosis, it is challenging to change the classification to include diagnostic categories that depend on genetype [33]. These challenges include the surrogate genotyping that may need to be taken by institutions without genotyping capabilities and the integrated diagnosis formats [33]. Thus, we aimed to show the possibility of automated classification in diffuse glioma, which may affect this complex format of diagnosis and molecular test platform, as well as facilitate routine initial diagnosis. Furthermore, the main hypothesis that is addressed in this work is that clinical-grade performance can be reached using ROI images without annotating WSIs. Furthermore, turnaround times for confirmatory molecular study are required up to a few weeks and they may result in a delay in diagnosis. The processing time of a slide using our model only takes a few seconds to calculate per classification probability on two NVIDIA Tesla V100 GPUs. When considering the possibility of using multiple GPUs to process patches in parallel, classification using our model can be executed in a few minutes.

The implications of these results are wide ranging. The possibility of the dataset without annotation allows our algorithm to learn from the full files of slides that are presented to clinicians from real-world clinical practice, representing the full wealth of biological and technical variablitiy [34]. To the best of our knowledge, only robust studies were conducted with the deep learning approach using the publicly available digital WSI dataset in The Cancer Genome Atlas (TCGA) or The Cancer Imaging Archive (TCIA) to automate the classification of grade II, III glioma versus grade IV glioblastoma, which demonstrated up to 96% accuracy [35,36,37] However, the datasets used in the above studies are composed of many cases diagnosed before the application of the WHO’s new 2016 classification; therefore, the algorithm that was developed using the public database might not be suitable for the current WHO classification system. A recent study tried deep learning approaches for subtype classification according to the 2016 WHO classification and survival prediction using multimodal magnetic resonance images of a brain tumor [38]. Their experimental data were obtained from the Multimodal Brain Tumor Segmentation Challenge 2019. However, they did not address the dataset’s information regarding molecular work to fulfill the requirement of the 2016 WHO classification and they did not include pathology images in their analysis. In order to generate predictive model using deep learning technique, sample size is important for classification performance and sample size. We included the dataset of diffuse glioma diagnosed after 2017 and all of the included cases were diagnosed according to the integrative molecular guideline of 2016 WHO classification. Interestingly, recent advanced biochemical spectroscopy enabled machine learning to discriminate between glioma and normal tissue by observing spectra shift within fresh tissue biopsies [39]. Such algorithms lay a potential scenario in performing biomolecular diagnosis at frozen diagnosis during surgery. The task of classifying the type and grade of glioma uses pre-defined image features, characterizing the image and predicting the classification level, which is not unlike other types of machine learning problem where the substantial disadvantage of pre-defined features is the need to know those that are most informative in the classification task. Often, the best features are hard to know, and a method of unsupervised feature learning can be advantageous if datasets were abundant [35].

The results of our evaluation of accuracy for ODG classification on real world data (87%) is promising, but it leaves room for improvement. There are several potential reasons for this performance. First, the orgin dataset of data set are HE slides and they vary in terms of coming from multiple institutions. tissue processing protocol, staining, and image acquisition were not uniform, which can bias the performance estimates of predictive models [40]. Second, the dataset of this cohort is imbalanced and each training set is small. Transfer learning is novel deep learning method that transfers pre-trained models from large datasets to new domains of interest with small datasets achieving good performance [41,42].

The accuracy for grading of diffuse glioma dataset (0.68%) is a reasonably preliminary result. The reason might be related to the difficulties in the interpretation of histologic criteria used to classify and grade the diffuse gliomas [43]. The main change in the 2016 WHO classification for diffuse glioma is providing powerful prognostic information from molecular parameters. The histologic grading system for diffuse glioma in current WHO scheme is three-tiered. Grade II is defined as tumors with cytological atypia alone. Grade III is considered to be tumors showing anaplasia and mitotic activity. Grade IV tumors show microvascular proliferation and/or necrosis. A variation in nuclear shape and size with accompanying hyperchromasia is defined as atypia. Unequivocal mitoses but not significant in their number or morphology are required. The finding of solitary mitosis is not sufficient for grade III, in such case, additional MIB1 proliferation index can be useful for the grading. Microvascular proliferation is defined as the multilayering of endothelium or glomeruloid vasculature. Any type of necrosis may enough for grade IV; palisading, simple apposition of cellular zone with intervening palor [44]. Thus, the crucial diagnostic component of WHO grading scheme is not always diffuse, but rather focal microscopic finding, and important information for tumor grading cannot be evenly included in patches made from the original pathology image. We expect that larger sample will improve the accuracy of our module.

For the application of deep learning algorithms in pathology image analysis, CNN based models are widely used for classification [45,46] and analysis, including the detection of tumor [47] and metastasis [48]. CNN can have a series of convolutional and pooling hidden layers [49]. This structure enables the extraction of representative features for prediction. Because the number of parameters is determined by the size of reception field, CNN layers have fewer parameters than the image size, which greatly improves its computational performance [50]. In the case of the CNN-based models pre-trained on ImageNet, which consists of about 10 million images that are usually used for a base model of transfer learning, the final feature vector produced by a neural network through serial convolutions often encodes redundant information, given the flexibility of the algorithm to choose any feature necessary to produce accurate classification. CNN have also been implemented for image segmentation [15]. Image segmentation is important in performing for large data sets, such as WSI. The image is divided into many small patches. CNN are trained to classify these patches. All of the patches are combined into a segmented area. A fine spatial resolution of segmentation is achieved by small size patches. However, in the patholohy image processing, the patch size should be reasonably large enough to be classified accurately. Thus, a review of the representative patches by a pathologist and the determination of proper patch size is recommended [51]. In our study, the proper patch size was determined after a review of representative patch by pathologist. However, this demands a huge computational time and memory, which imits the computational speed. There is always an issue of the tradeoff between resolution of segmentation and patch size. We divide the 224 × 224 pixels sized patches from the original size image of 1024 × 1024 pixels to adjust the image size for ResNet50V2 model and show the best performance in our model.

The flexible adjustment of the deep learning process should be performed according to data size and trait, including data preparation, image processing, model selection and construction, post-processing, and feature extraction, as well as the association with the disease [15]. Class imbalance is a common problem that has been comprehensively studied in classical machine learning. The clinical dataset is inevitably unbalanced because the natural incidence of tumor types dependent on the prevalence of the tumor. Oversampling images to adjust the ratio of each images should especially be performed in the small dataset [52]. Because pathology images may look very different due to different hematoxylin and eosin staining condition and the thickness of original slide, it is important to make a deep learning algorithm that is adapted to different digital images. Image standardization is needed and color augmentation is the easiest solution among the reported methods [53]. Model selection is important and the corresponding loss function should be done. We performed training loss from four transfer learning models and four early-stopped validation losses for each model. Our study was performed as an annotation-free WSI training approach for pathological classification of the brain tumor subtype. Deep learning for digital pathology uses the extremely high spatial resolution of WSIs. The image patch usually used in learning needs detailed annotation. However, it is difficult for pathologists to cover all possible samples during annotation due to the highly variable tissue histology. Weak supervision methods have been applied in recent studies to avoid the work burden of annotation and selection bias. Training the tumor classifier without annotations in detail reduces the burden on the expert pathologist and allows for the deep learning model to benefit from abundant readily available WSIs [54].

Our algorithm lets us meet the minimum requirement of the accuracy level that is determined by the context of complex molecular and histologic determination within a multi-class classification scheme. Moreover, as far as we know, our work is the first to get closer to actual practice by exploring how deep transfer learning methods can be used to classify brain tumor according to the new integrated diagnosis of the 2016 WHO classification. There are many factors of deep transfer learning optimization that we have not yet explored. We will also work on improving our accuracy by extra-steps during the pre-processing stage.

The era of big data has produced vast amounts of information that can be used to build deep learning models. However, in many cases, adding more data only marginally increases the model performance. This is especially important for limited labeled data, as the process annotation can be expensive and time consuming. The evaluation of learning curve approximation for large imbalanced biomedical datasets in the context of sample size planning can provide guidance for future machine learning problems that require expensive human labeling of instances. [55]. On the basis of a systematic study on the sample size prediction for classification [56], learning curves can be represented using inverse power law functions. By utilizing this, a classifier’s accuracy $Y_{\mathrm{acc}}\left(n\right)$ can be expressed as a function of the number of training examples:$$Y_{\mathrm{acc}}\left(n\right)=\left(1-a\right)-r\cdot n^{d}$$
where *a* is the minimum achievable error, *r* is the learning rate, and *d* is the decay rate.

Becaue we aimed for an accuracy of 90% and used the learning rate of 0.001 and the decay rate of −0.1, we could identify the trend of the accuracy, as shown in Figure 5. From the above accuracy trend, we estimate 10,000 to be the minimum required sample size, because the increase of accuracy is very small after 10,000. Sample number determination is important for model performance, especially in the context of limited labeled data [55]. However, our ROI image selection and quality control of WSI during routine clinical practice generate a high quality dataset that showed fair performance. Clinically performed quality control could be an alternative to vigorous annotation.

## 5. Conclusions

In summary, our preliminary study provided a proof of concept for incorporating automated glioma subtype screen using routine WSI into the pathology workflow to reduce the cost of expensive genetic test and, furthermore, to augment the productivity of the pathologist. Future studies will need to include a far more balanced dataset of WSI, and more cases for training, validation, and testing. Our preliminary results highlight the usability of quality controlled clinical grade data and the possible role of deep learning in precision medicine, and suggest an expanding utility for computational analysis of pathology in future clinical practice.

## Figures and Tables

**Figure 1 sensors-21-03500-f001:**
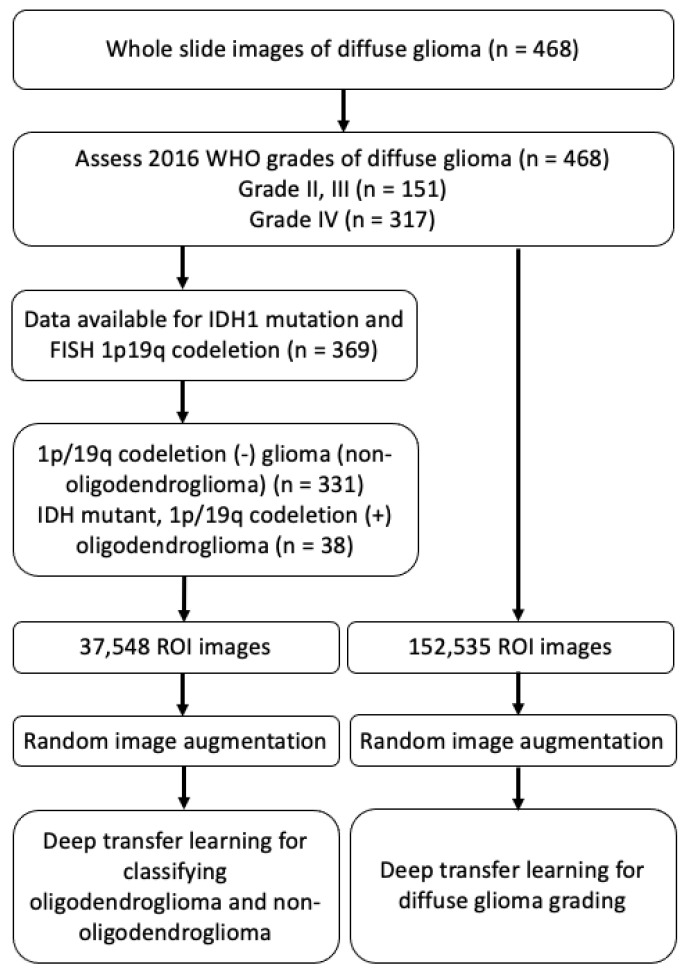
Schematic illustration of the study design and dataset selection. ROI; region of interest.

**Figure 2 sensors-21-03500-f002:**
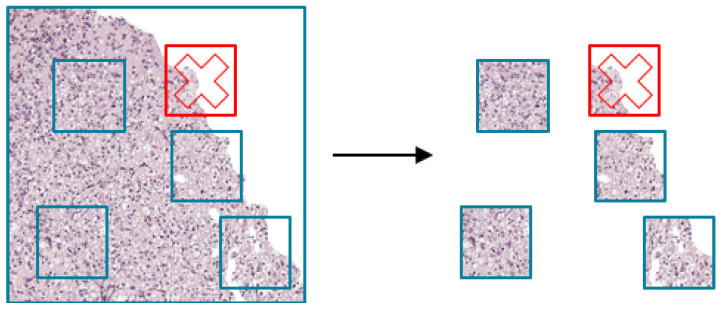
Random patch selection from region of interest images. The patch that wsa filled with more than 50% of white background was discarded (red cross).

**Figure 3 sensors-21-03500-f003:**
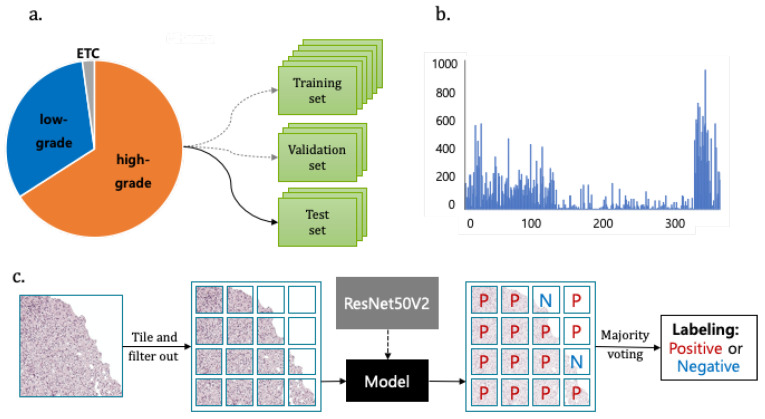
The dataset and strategy for the classification of glioma subtype using deep transfer learning and majority voting technique (**a**). The ratio of low-grade vs. high grade gliomas. (**b**) The distribution of patch numbers from one Whole Slide Image. (**c**) Transfer learning with ResNet50V2 model and majority voting technique for labeling glioma patient images.

**Figure 4 sensors-21-03500-f004:**
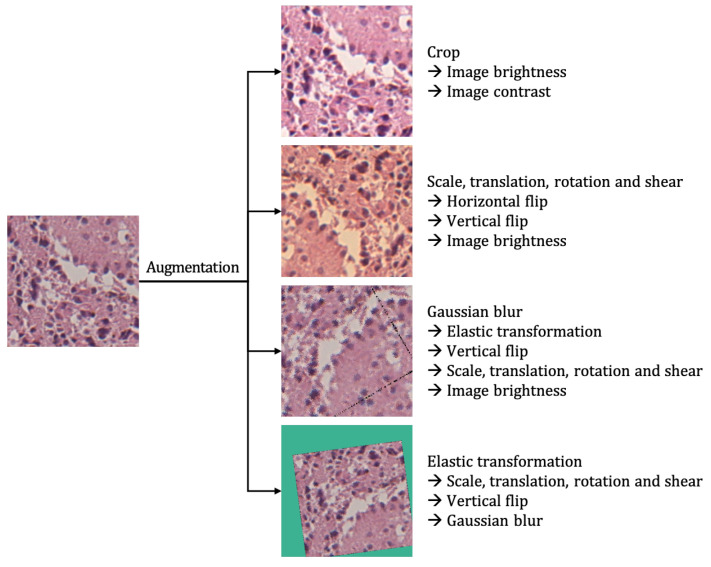
Application of random image augmentation to an image patch. The augmentations applied to the patch are listed on the right.

**Figure 5 sensors-21-03500-f005:**
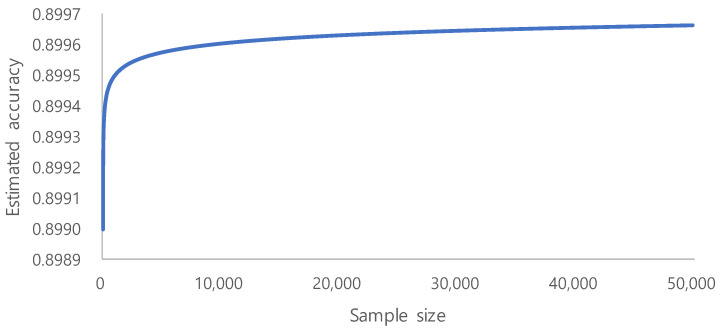
The trend of the accuracy as the number of training examples increases.

**Table 1 sensors-21-03500-t001:** The dataset of diffuse glioma.

Label	No. of Patients	No. of ROI Images
**Non-ODG glioma**	331	24,418
**ODG**	38	13,130
**Grade II–III**	148	46,349
**Grade IV**	320	106,187

ODG, oligodendroglioma; ROI, region of interest.

**Table 2 sensors-21-03500-t002:** Dataset for classifying non-ODG and ODGs gliomas.

Label	Training (N = 132)	Validation (N = 19)	Testing (N = 219)
	No. of ROIs	No. of ROIs	No. of ROIs
Non-ODG glioma	17,580 (N = 113)	1954 (N = 16)	4884 (N = 203)
ODG	9461 (N = 19)	1052 (N = 3)	2617 (N = 16)

ODG, oligodendroglioma; ROI, region of interest.

**Table 3 sensors-21-03500-t003:** The dataset for grading diffuse gliomas.

Label	*Training (N = 293)*	*Validation (N = 81)*	*Testing (N = 94)*
	No. of ROIs	No. of ROIs	No. of ROIs
**Glioma, grade II, III**	29,663 (N = 92)	7416 (N = 29)	9270 (N = 27)
**Glioma, grade IV**	67,959 (N = 201)	16,990 (N = 52)	21,238 (N = 67)

ROI, region of interest.

**Table 4 sensors-21-03500-t004:** The proportion of random image augmentation.

Probability	Augmentation Technique
50%	Horizontal flip
50%	Vertical flip
30%	Crop
30%	Scale, translation, rotation and shear
30%	Gaussian blur
30%	Image contrast
30%	Gaussian noise
30%	Image brightness
30%	Elastic transformation

**Table 5 sensors-21-03500-t005:** The early-stopped validation loss for each model.

Model	Epoch	Loss
**EfficientNet-B4**	3	0.6432
**EfficientNet-B5**	3	0.6586
**DenseNet201**	6	0.6646
**MnasNet**	18	0.6663

**Table 6 sensors-21-03500-t006:** The training loss from four transfer learning models.

Model	Loss
**ResNet50V2**	0.4086
**InceptionV4**	0.5109
**Xception**	0.4213
**DenseNet201**	0.5193

**Table 7 sensors-21-03500-t007:** The ODG image classification model performance without and with majority voting.

Metric	Performance	Performance with Majority Voting
**Precision**	0.7346	0.2778
**Recall**	0.7119	0.9375
**F1**	0.7231	0.4286
**Accuracy**	0.8098	0.8174
**Balanced Accuracy**	0.7870	0.8727

**Table 8 sensors-21-03500-t008:** The performance of the diffuse glioma grading model.

Model	Accuracy	Precision	Recall	Balanced Accuracy
**EfficientNet-B4**	0.6219	0.3752	0.3672	0.5501
**EfficientNet-B5**	0.5957	0.3823	0.5366	0.5791
**DenseNet201**	0.6810	0.4591	0.2790	0.5678
**MnasNet**	0.6389	0.4102	0.4300	0.5801

## Data Availability

The datasets generated during and/or analysed during the current study are available from the corresponding author on reasonable request.

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
