# Peer review of "Classification of Diffuse Glioma Subtype from Clinical-Grade Pathological Images Using Deep Transfer Learning"

_sensors, 2021, doi:10.3390/s21103500_

Round 1

Reviewer 1 Report

This is an interesting area of research and the authors have collected a unique dataset using cutting edge methodology. The paper is generally well written but not well structured. However, in my opinion the paper has lots of shortcomings in regards to study design,  data analyses and text, and I feel this unique dataset has not been utilized to its full extent.

1. Flow charts are a very helpful method to understand this kind of structure which is missing.

2. The sample distribution is completely random. For example: section 2.2: training N=132, validation N=18, and testing N=219 with evidence of selection criteria. Same trend followed throughout the manuscript.

3. Sample size/power calculation is missing.    

4. Why did the authors use two terms such as validation and test because both have played a  similar role? Please explain the difference?

5. The literature review was not thorough, following are some similar studies published recently but authors have not provided the any comparison to prove their method is much improved as compared to other published approaches:
  • Optimization of deep learning methods for visualization of tumor heterogeneity and brain tumor grading through digital pathology (Neuro-Oncology Advances, https://doi.org/10.1093/noajnl/vdaa110)
  • Segmentation and Classification in Digital Pathology for Glioma Research: Challenges and Deep Learning Approaches (Front. Neuroscience, PMID: 32153349).
  • Context aware deep learning for brain tumor segmentation,subtype classification, and survival prediction using radiology images (Scientific Reports, PMID: 33184301)

Author Response

This is an interesting area of research and the authors have collected a unique dataset using cutting edge methodology. The paper is generally well written but not well structured. However, in my opinion the paper has lots of shortcomings in regards to study design,  data analyses and text, and I feel this unique dataset has not been utilized to its full extent.

1. Flow charts are a very helpful method to understand this kind of structure which is missing.

Thank you for the precious comment.

- We revised the detailed description of study design and dataset in material and method and added flow chart as Figure 1.

- We revised material and method as “All cases had pathology data including the diagnosis and grades (grade II, III and IV) according to 2016 WHO classification. Among them, full molecular profiles could be assessed in 369 cases including IDH1 mutation status tested by immunohistochemistry, and 1p/19q co-deletion status examined by FISH. Based on above molecular result, WSIs from IDH mutant, 1p/19q co-deleted ODGs (N=38) and 1p/19q codeletion (-) diffuse glioma subtypes (non-ODG glioma) (N$=331) were used for developing ODG and non-ODG glioma binary classification model. And WSIs from 468 diffuse glioma dataset were used in binary classification of diffuse gliomas into grade II-III and grade IV. Each model was performed separately.The clinical information was obtained from electric medical record.”

- We revised detailed information regarding WSI images, ROI selection and cropping patches as “The whole slide digital pathology images were generated during performing 1p/19q FISH test using the automated BioView Duet scanning system (BioView, Rehovot, Israel). Images from diffuse glioma cohorts were reviewed to removed images having tissue process artifacts, including section fold, pen mark, poor staining and bubbles. Representative region of interests (ROIs) containing brain tumor were manually selected for each WSI that passed a quality control review during routine clinical practice. 37,548 ROI images and 152,535 ROI images are prepared for Deep transfer learning of classification of ODG and diffuse glioma grading, respectively.”

2. The” sample distribution is completely random. For example: section 2.2: training N=132, validation N=18, and testing N=219 with evidence of selection criteria. Same trend followed throughout the manuscript

-

Thank you for the precious comment. We divided the samples into three different groups (training, validation and testing) as 7:1:2 ratio based on the number of region of interest(ROI) images not the number of patients as shown in Table 2 and Table3. We also revised section 2.2 and Table 2 and Table 3 to improve readability.

3. Sample size/power calculation is missing.    

Thank you for the precious comment. We added learning curve using inverse power law function described in previous study regarding sample size requirement estimation for classification and added Figure 5 and revised discussion as “On the basis of a systematic study on sample size prediction for classification {figueroa2012predicting}, learning curves can be represented using inverse power law functions.

And we addressed “10,000 as the minimum required sample size because the increase of accuracy is very small after 10,000. However, In viewpoint of sample number in limited labeled data, our ROI image selection and quality control of WSI during routine clinical practice can generate high quality data and could be an alternative for vigorous annotation.”

4. Why did the authors use two terms such as validation and test because both have played a  similar role? Please explain the difference?

- Thank you for the important comment. We utilized the validation set for parameter tuning and the test set for performance measurement of our proposed classification model and referenced above method in our material and method.

5. The literature review was not thorough, following are some similar studies published recently but authors have not provided the any comparison to prove their method is much improved as compared to other published approaches:

  • Optimization of deep learning methods for visualization of tumor heterogeneity and brain tumor grading through digital pathology (Neuro-Oncology Advances, https://doi.org/10.1093/noajnl/vdaa110)
  • Segmentation and Classification in Digital Pathology for Glioma Research: Challenges and Deep Learning Approaches (Front. Neuroscience, PMID: 32153349).
  • Context aware deep learning for brain tumor segmentation,subtype classification, and survival prediction using radiology images (Scientific Reports, PMID: 33184301)

Thank you for the precious comment.

We included above references in our manuscript.

  1. As for the first two references which used TCGA public data including many cases diagnosed before 2016 WHO classification, we included them in discussion as “To the best of our knowledge, only robust studies were conducted with deep learning approach using publicly available digital WSI dataset in The Cancer Genome Atlas (TCGA) or The Cancer Imaging Archive (TCIA) to automated classification grade II, III glioma versus grade IV glioblastoma demonstrated up to 96% accuracy {ertosun2015automated, truong2020optimization, kurc2020segmentation} However, the datasets of used in above studies are composed of many cases diagnosed before the application of new 2016 WHO classification, therefore, the algorithm developed using public database might not be suitable for the current WHO classification system.”
  2. As for the third reference, they used only radiology images (mMRI) and they did not address the data obtained from Brain tumor classification competition having the information regarding integrative molecular diagnosis according to 2016 WHO classification. So, we included the reference as “Recent study tried deep learning approaches for subtype classification according to 2016 WHO classification and survival prediction using multimodal magnetic resonance images of brain tumor {pei2020context}. However, they did not address the dataset's information about molecular work to fulfill the requirement for 2016 WHO classification and they did not include pathology images in their analysis.

Reviewer 2 Report

The paper describes the development of a deep learning approach to automatically classify glioma subtypes. In particular, a deep transfer learning method with the ResNet50V2 model is used to classify glioma subtypes from 369 whole-slide images acquired in routine practice. It is also proposed a method for grading diffuse glioma according to the guidelines released by WHO in 2016. According to the authors (line 56), this is the first study conducted to classifying diffuse gliomas using real-world pathology images.

The use of transfer learning in this study is particularly useful to adapt a model trained on a great amount of data to the field of interest for which only a smaller dataset is available. As stated by the authors the results obtained are still partial. Despite this, they provide some evidence that the proposed method is innovative and may pave the way for the use of deep learning in the field of pathology images to classify brain tumors. The provided references are up to date, figures and tables are exhaustive. For these reasons, the paper can be considered in line with the aim of the journal, I recommend acceptance after revision.

Major issues

I would suggest the authors move lines 156-163 from section 3 to section 2. Also, the technical details could be further expanded and clarified to ensure that readers understand exactly what the researchers studied.

The literature survey should be expanded adding recent work on the topic (e.g., Riva, Marco, et al. "Glioma biopsies Classification Using Raman Spectroscopy and Machine Learning Models on Fresh Tissue Samples." Cancers 13.5 (2021): 1073.)

In the discussion section, it is stated for the first time that the dataset is not annotated. I would suggest the authors add this information in the section where the dataset is described and then mention it again in this part to highlight its utility.

Minor issues

Please make sure that all the meanings of the abbreviations are given the first time that they are mentioned, to give a better understanding to the reader.

Please consider adding a description or a citation for the method named in line 81.

In line 111 are listed the architectures used to develop the model. Please consider citing all of them.

In the caption of figure 2, it could be useful to define to which augmentation technique corresponds each part of the image.

The authors could clarify the sentences in lines 151-155 and 219-220 to avoid confusion and improve readability.

Some words are misspelled, please consider revise them to improve understandability.

Author Response

Thank you for the precious recommendations. According to the feedback from the Reviewer, we revised introduction, methods, and results in our manuscript to improve readability. We added flow charts as figure 1 to explain research design and methods,

Major issues

  1. I would suggest the authors move lines 156-163 from section 3 to section 2. Also, the technical details could be further expanded and clarified to ensure that readers understand exactly what the researchers studied.

Thank you for the precious comment. We moved lines 156-163 to section 2.5, which is the part of transfer learning process. Also, we updated section 3.2 with more detailed references and updated Table 6. 

  1. The literature survey should be expanded adding recent work on the topic (e.g., Riva, Marco, et al. "Glioma biopsies Classification Using Raman Spectroscopy and Machine Learning Models on Fresh Tissue Samples." Cancers 13.5 (2021): 1073.)

Thank you for the good reference, which used biochemical biopsy using spectroscopy, which is the very updated method for diagnostic field. So, we included above reference in our discussion as” Interestingly, Recent advanced biochemical spectroscopy enabled machine learning to discriminate between glioma and normal tissue by observing spectra shift within fresh tissue biopsies {Riva2021Glioma}. Such a machine learning algorithms lay a potential scenario to performing biomolecular diagnosis at frozen diagnosis during surgery.”

  1. In the discussion section, it is stated for the first time that the dataset is not annotated. I would suggest the authors add this information in the section where the dataset is described and then mention it again in this part to highlight its utility.

Thank you for the important comment. The selection of ROI images from WSI minor image curation is a part of routine clinical practice for the diagnosis. So, we used those selected images instead of annotation. We added revised and referenced such image selection information in our material and method section as” The whole slide digital pathology images were generated during performing 1p/19q FISH test using the automated BioView Duet scanning system (BioView, Rehovot, Israel). Images from diffuse glioma cohorts were reviewed to removed images having tissue process artifacts, including section fold, pen mark, poor staining and bubbles. Representative region of interests (ROIs) containing brain tumor were manually selected for each WSI that passed a quality control review} \cite{mobadersany2018predicting} during routine clinical practice. 37,548 ROI images and 152,535 ROI images are prepared for Deep transfer learning of classification of ODG and diffuse glioma grading, respectively. \hl{The study design and WSI dataset selection are illustrated in Fig1

Minor issues

  1. Please make sure that all the meanings of the abbreviations are given the first time that they are mentioned, to give a better understanding to the reader.

Thank you for the important comment. We have checked abbreviations are given the meanings at the first time they are mentioned in manuscript, figure legends and tables.

  1. Please consider adding a description or a citation for the method named in line 81.

Thank you for the important comment. We added the references for the knapsack algorithm on lines 100.

  1. In line 111 are listed the architectures used to develop the model. Please consider citing all of them.

Thank you for the comment. We added the references to all four models. 

  1. In the caption of figure 2, it could be useful to define to which augmentation technique corresponds each part of the image.

Thank you for the comment. As the reviewer suggested, we revised the figure 3 to describe which augmentation methods were applied for each patch.

The authors could clarify the sentences in lines 151-155 and 219-220 to avoid confusion and improve readability.

Thank you for the comment. As recommended, we revised the sentences in lines 151-155 and 219-220 to improve readability.

Some words are misspelled, please consider revise them to improve understandability.

Thank you for the comment. We have checked and revised the typos and highlighted them.

Reviewer 3 Report

This paper considers the task of classification of dffuse glioma subtype from clinical-grade pathological images using transfer learning. Totally speaking, this paper does not present as a scientific research paper. The exploited method does not show sufficient novelty, which merely exploited an off the shelf transfer learning framework for classification. Additionally, from the perspective of application, no special domain challenge is shown and addressed.

Author Response

Thank you for the precious comment. as reviewer suggested, we revised the manuscript and updated the introduction, research design, methods, and results to improve readability.

1) The important point of our study is the first try of using real world dataset for glioma classification which were clinically generated. We performed the quality-controlled image selection of region of interest, and image curation during clinical diagnosis using digital pathology system. Without additional annotation, we demonstrated promising result using clinical grade dataset even though shortcomings in sample size and study design. We revised the detailed description of study design and dataset in material and method and added flow chart as Figure 1.

2)         We revised material and method as “All cases had pathology data including the diagnosis and grades (grade II, III and IV) according to 2016 WHO classification. Among them, full molecular profiles could be assessed in 369 cases including IDH1 mutation status tested by immunohistochemistry, and 1p/19q co-deletion status examined by FISH. Based on above molecular result, WSIs from IDH mutant, 1p/19q co-deleted ODGs (N=38) and 1p/19q codeletion (-) diffuse glioma subtypes (non-ODG glioma) (N$=331) were used for developing ODG and non-ODG glioma binary classification model. And WSIs from 468 diffuse glioma dataset were used in binary classification of diffuse gliomas into grade II$-$III and grade IV. Each model was performed separately.The clinical information was obtained from electric medical record.”

3)  We revised detailed information regarding WSI images, ROI selection and cropping patches as “The whole slide digital pathology images were generated during performing 1p/19q FISH test using the automated BioView Duet scanning system (BioView, Rehovot, Israel). Images from diffuse glioma cohorts were reviwed to removed images having tissue process artifacts, including section fold, pen mark, poor staining and bubbles. Representative region of interest (ROI) images containing brain tumor were manually selected for each WSI that passed a quality control review during routine clinical practice. 37,548 ROI images and 152,535 ROI images are prepared for Deep transfer learning of classification of ODG and diffuse glioma grading, respectively.

4) We added sample number determination using learning curve as described previous sample size prediction for classification and added the result in discussion.

And we compared the importance of sample number in unlabeled database with our data that However, our ROI image selection and quality control of WSI during routine clinical practice generate high quality dataset and showed fair performance. Clinically performed quality control could be an alternative for vigorous annotation

Round 2

Reviewer 3 Report

I have no more comments